# Systematic Review of Nicotine Exposure’s Effects on Neural Stem and Progenitor Cells

**DOI:** 10.3390/brainsci11020172

**Published:** 2021-01-29

**Authors:** Arrin C. Brooks, Brandon J. Henderson

**Affiliations:** Department of Biomedical Science, Joan C Edwards School of Medicine, Marshall University, Huntington, WV 25545, USA; hendersonbr@marshall.edu

**Keywords:** nicotine, neural stem cells, neurogenesis, subventricular zone

## Abstract

While various modalities of chronic nicotine use have been associated with numerous negative consequences to human health, one possible benefit of nicotine exposure has been uncovered. The discovery of an inverse correlation between smoking and Parkinson’s disease, and later Alzheimer’s disease as well, motivated investigation of nicotine as a neuroprotective agent. Some studies have demonstrated that nicotine elicits improvements in cognitive function. The hippocampus, along with the subventricular zone (SVZ), is a distinct brain region that allow for ongoing postnatal neurogenesis throughout adulthood and plays a major role in certain cognitive behaviors like learning and memory. Therefore, one hypothesis underlying nicotine-induced neuroprotection is possible effects on neural stem cells and neural precursor cells. On the other hand, nicotine withdrawal frequently leads to cognitive impairments, particularly in hippocampal-dependent behaviors, possibly suggesting an impairment of hippocampal neurogenesis with nicotine exposure. This review discusses the current body of evidence on nicotine’s effects on neural stem cells and neural progenitors. Changes in neural stem cell proliferation, survival, intracellular dynamics, and differentiation following acute and chronic nicotine exposure are examined.

## 1. Introduction

Neurogenesis in the adult mammalian brain primarily occurs in two distinct regions: the subventricular zone (SVZ) located along the walls of the lateral ventricles and the subgranular zone (SGZ) located in the dentate gyrus (DG) of the hippocampus. Neural stem cells (NSCs) reside and continue to symmetrically divide throughout adulthood in these regions giving rise to transiently amplifying multipotent neural progenitor cells (NPCs) [1,2,3]. Neurogenesis in the DG plays a particularly important role in hippocampal-mediated learning [4,5]. In fact, enhanced neurogenesis and survival of newborn cells in the hippocampus is observed in rats trained on hippocampal-dependent tasks [5], and performance on these tasks is impaired by inhibition of cell proliferation [6]. Newly generated NPCs in the SVZ give rise to another type of neural precursor called the migrating neuroblast, which travel from the lateral ventricles to the olfactory bulb (OB) along the rostral migratory stream (RMS) where they then mature into neural cells involved in olfaction [7,8,9,10,11,12].

Because both clinical and preclinical studies have demonstrated that nicotine elicits improvements in cognitive function [13,14,15,16,17,18,19] and possibly offers neuroprotection against neurodegenerative conditions [20,21], many studies have sought to elucidate the effect of nicotine on neuroplasticity. In contrast to these studies driven by the hypothesis that nicotine is a neuroprotective compound [22,23,24], others have sought to evaluate nicotine’s effects on stem cells. This is due to evidence that nicotine is a neuroteratogen and alters blood-brain barrier (BBB) function [25,26,27] and potentially contributes toward negative consequences for NSC health. Further, studies have shown abstinence from smoking in nicotine-dependent individuals leads to profound cognitive impairment [28,29,30] and disruption of hippocampal and prefrontal cortex (PFC) associated behaviors [31,32,33,34,35], suggesting that chronic nicotine exposure might impair or otherwise alter mechanisms related to learning and memory such as hippocampal neurogenesis.

While the research of possible nicotine-related effects on NSC survival, proliferation and differentiation is understudied, investigation into this area has highly important clinical implications. Although smoking rates have declined in recent decades, the 32nd tobacco-related Surgeon General’s report states that 42 million American adults and about 3 million middle and high school students continue to smoke. Additionally, nicotine use in the form of electronic nicotine delivery systems (ENDS) (also known as e-cigarettes (ECs)) has substantially gained popularity. Recent studies report that youth who use ENDS may have a higher propensity to smoke combustible cigarettes in later life [36]. In 2019, over 5 million U.S. middle and high school students had reported ENDS use in the past 30 days, including 10.5% of middle school students and 27.5% of high school students [37]. As brain development continues into the mid-twenties, elucidation of the neurological consequences of nicotine exposure is paramount [38,39].

## 2. Nicotine-Related Decreases in NSC Proliferation

An in vitro study showed prolonged exposure to nicotine decreased proliferation of NSCs isolated from embryonic whole brain [40] (Table 1). Brains collected from embryonic day 16 mice were enzymatically treated to isolate NSCs and were cultured as neurospheres. Neurospheres were then treated with vehicle or nicotine at 100, 400, and 800 μM doses, concentrations that greatly exceed that of what is clinically relevant, for five days. In a physiological setting, nicotine in the brain of combustible cigarette and ENDS smokers is typically between 0.05 and 0.5 μM [41]. The proliferative capacity of nicotine treated NSCs was then analyzed via quantity and size measurements of the cultured neurospheres. Nicotine decreased the number and size of newly formed neurospheres as compared to vehicle. Both diminished size and number were dose-dependent and could be restored by application of mecamylamine (MECA), a nonselective nicotinic acetylcholine receptor (nAChR) noncompetitive antagonist. Using RT-PCR to measure mRNA expression of nestin, an NPC marker, and proliferating cell nuclear antigen (PCNA), it was shown that nicotine inhibits proliferation of neurospheres at a transcriptional level. In the RT-PCR analysis, the levels of nestin mRNA were not changed, whereas the level of PCNA was significantly decreased. Additionally, confocal microscopy revealed nicotine treatment increased the number of condensed nuclei and shortened neurogenesis in a dose-dependent manner compared to vehicle. Nicotine also increased the release of certain pro-inflammatory cytokines in NSCs. RT-PCR analysis exposed increased levels of cyclooxygenase-2 (COX-2), tumor necrosis factor alpha (TNF-α), and glutathione reductase mRNA in the nicotine-treated neurosphere cultures. Intriguingly, variable TNF-mediated signal cascades in NSCs have been identified. Depending on which receptor subtype is activated, NSC proliferation can be increased or decreased in response to TNF-α [42,43]. Autocrine signaling of TNF-α is suggested to play a role in the cytokine’s influence on neurogenesis [43,44]. In addition to increased transcription of these pro-inflammatory genes, nicotine was shown to increase the mRNA levels of histone deacetylase 1 (HDAC1) and decrease the mRNA levels of sirtuin 1 (SIRT 1, a nicotinamide adenine dinucleotide (NAD)-dependent histone and protein deacetylase) in NSCs. HDAC2 was not changed with nicotine treatment. Nicotine also upregulated levels of inducible nitric oxide synthase (iNOS). When HDAC inhibitors, sodium butyrate (NaB) or valproic acid (VPA), were added to NCS cultures for five days, HDAC1 mRNA was decreased. However, neither NaB nor VAP changed mRNA levels of HDAC2 or SIRT1. Furthermore, HDAC inhibitors partially rescued proliferation of NSCs compared to nicotine without NaB or VPA.

These results are interesting in light of other work that has reported chronic nicotine exposure to decrease HDAC activity in mice [45]. Histone acetylation is an epigenetic modification to favor gene expression that is moderated by opposing actions of histone acetyltransferase (HAT) and HDAC, adding and removing acetyl groups, respectively. Therefore, decreased HDAC activity would enhance histone acetylation and promote gene expression. Commonly, activation of HDAC1 would result in transcriptional repression of many gene targets. In the aforementioned study, it is speculated that HDAC1 might deacetylate a repressor bound to an iNOS promoter [46].

Epigenetic regulation has emerged in the literature as an important mechanism underlying brain changes associated with nicotine and other drugs of abuse exposure. In conjunction with the described study, others have demonstrated nicotine to decrease SIRT1 levels in cultured mouse embryos [47]. Sirtuins are a family of NAD-dependent enzymes thought to be involved in cell metabolism and regulation of DNA repair, inflammatory response, cell cycle, and apoptosis [48]. Interestingly, SIRT1, the sirtuin shown to decrease following nicotine exposure, demonstrates a general neuroprotective effect. In fact, hypermethylation of the SIRT1 gene and significant decrease of SIRT1 expression has been observed in patients with Alzheimer disease [49]. Furthermore, epigenetic mechanisms were confirmed in vitro [50]. These and other nicotine-induced epigenetic changes could be responsible for the negative effects of nicotine on NSC proliferation.

Nicotine impairment of neural plasticity has also been observed in multiple in vivo studies [51,52,53]. Abrous et al., found that nicotine self-administration greatly decreased expression of polysialylated (PSA) forms of neural cell adhesion molecule (NCAM) in the DG in a dose-dependent manner [51]. They also observed nicotine-related decreases in neurogenesis and increased cell death (Figure 1). Using male Sprague Dawley rats and an intravenous self-administration (IVSA) model, animals self-administer one of three unitary doses of nicotine (0.02, 0.04, and 0.08 mg/kg per infusion) or vehicle solution, delivered on a contingent basis. Following IVSA, brain sections were collected and analyzed for PSA-NCAM and Brdu (Bromodeoxyuridine/5-bromo-2′-deoxyridine) immunoreactivity. PSA-NCAM was evaluated because in the adult hippocampus PSA-NCAM is expressed in newborn neurons [54] and in mutant mice, modifications of PSA-NCAM expression causes morphological changes to these hippocampal cells and worsens cognitive function [55]. PSA-NCAM immunoreactive cells were located in the deepest region of the granule cell layer at the interface of the hilus within the hippocampus. Nicotine IVSA diminished PSA-NCAM expression in the DG, a decrease up to 44% less than control in the 0.04 mg/kg nicotine dose (Figure 1). Notably, the PSA-NCAM expression was unaltered in the SVZ of nicotine-treated rats. Immunoreactivity for Brdu was decreased in the DG of rats that received nicotine IVSA compared to those receiving vehicle. Again, Brdu-labeling was not altered with nicotine IVSA in the SVZ. Both changes in PSA-NCAM immunoreactivity and Brdu-labeling were most significantly modified by the medium (0.04 mg/kg) dose of nicotine. Evaluation of either neuronal cell marker, NeuN, or astrocytic marker, GFAP, were co-labeled with Brdu to reveal the phenotype of proliferating cells. In control animals, approximately 7% of Brdu-stained cells expressed astroglial marker GFAP, and about 60% of Brdu-stained cells were NeuN-positive. In nicotine IVSA rats, the percentage of Brdu-stained cells co-labeled with GFAP or NeuN was not significantly different than the percentages of that in control animals. While the total number of Brdu/GFAP-positive cells was also not changed by nicotine administration, the total number of Brdu/NeuN-positive cells was decreased in a dose-dependent manner. The medium dose of nicotine (0.04 mg/kg) had the least amount of Brdu-stained NeuN-positive cells, equaling about half the amount as the Brud/NeuN-positive cells in control. Nicotine IVSA also resulted in greater cell death within the granule cell layer of the DG, demonstrating a significantly increased number of pyknotic cells with dose-dependent nicotine exposure in this region. Again, the greatest number of pyknotic cells within the DG was actually seen in the medium dose of nicotine, with both 0.04 and 0.08 mg/kg nicotine doses resulting in heightened cell death as compared to control or lower dose (0.02 mg/kg) nicotine.

In response to the seemingly controversial work of Abrous et al., as their study appeared inconsistent with prior work that had reported cognitive-enhancing properties of nicotine [15,16,56,57,58], Scerri et al., sought to investigate the effects of constantly infused nicotine on rat spatial learning in the Morris water maze and cell proliferation in the DG [52]. In this study, two doses of nicotine were chosen, the high dose (4 mg/kg daily) representing blood nicotine concentrations (approximately 80 ng/mL) found in heavy smokers and the lower dose (0.25 mg/kg daily) to approximate concentrations (approximately 8 ng/mL) commonly found in the plasma of light smokers. They found that only the high dose nicotine (HDN) animals, and not the low dose nicotine (LDN) animals, took longer than controls to find the platform. Moreover, in probe trial, only HDN rats spent significantly less time than control rats in the region where the platform had been located. The group difference was not attributed to changes in swim speed. In all groups (HDN, LDN and control), spatial learning increased the number of Brdu+ cells in the DG as compared to rats not receiving the learning task. Administration of HDN reduced the number of Brdu+ cells in both trained and non-trained rats. Further, HDN, but not LDN, decreased the number of cells labelled with Brdu compared to controls.

One other study observed nicotine-related decreases in hippocampal neurogenesis [53]. Interestingly, Shingo et al., had shown in a prior study that nicotine administration in rats increased mRNA expression of insulin-like growth factor 1 (IGF1) within the hippocampus and cerebral cortex. Here, the authors concluded that this was partially responsible for nicotine’s beneficial effects on hippocampal and cortical neurons [59]. In their later study, nicotine was injected intraperitoneally (i.p.) (doses 0.1, 0.5 or 1 mg/kg) into adult male Wistar rats for two weeks and immunohistochemistry performed on hippocampal tissue slices to label PSA-NCAM, NeuN and GFAP [53]. The quantity of PSA-NCAM+ and NeuN+ cells were decreased greatly by nicotine (Figure 1). While the actual count of immunolabeled cells was not provided, the values expressed as percent of control revealed a dose-dependent decrease of PSA-NCAM- and NeuN-labelled cells in the DG with nicotine exposure (Figure 1). The highest dose of nicotine used (1 mg/kg) had a nearly 80% decrease in both PSA-NCAM+ and NeuN+ cells, while the percentage of control for GFAP-labelling was unchanged. Although the 0.5 mg/kg nicotine dose alone did reach a statistically significant increase in GFAP-labelling, the increase was not as great a percentage change as that found in PSA-NCAM or NeuN labelling, and the authors thus reported nicotine has no effects of the number of GFAP+ cells. Moreover, Shingo et al., suggest that nicotine may have a “two-sided effect” on neurons, having a positive effect on mature neurons and a negative effect on developing neurons (Figure 1).

## 3. Nicotine-Related Increases in NSC Proliferation

In contrast to studies mentioned above that have demonstrated either decreased neural precursor cells number within the hippocampus or no change in NSC number within the SVZ, Mudò et al., reported that acute intermittent nicotine treatment enhanced neural precursor proliferation in the SVZ [60] (Table 2). Adult male Wistar rats were administered a total of four injections given i.p. every 30 min at a dose of 1 mg/kg for the acute intermittent nicotine treatment group and similarly treated with saline for the control group. In order to evaluate the effect of nicotine on precursor cell proliferation, i.p. injections of Brdu 2 h before sacrifice were also used to label proliferating cells. Changes in neural precursor cell proliferation were measured at 24, 36, 48, and 72 h after acute intermittent nicotine treatments. The time-course study revealed a significant enhancement of SVZ precursor cell proliferation as seen by increased number of Brdu+ cells in the SVZ of nicotine treated rats versus controls at 36 h after nicotine exposure and maintained at least 72 h after [60]. Nicotine did not induce neural precursor proliferation in the SGZ of the adult hippocampus. The enhanced SVZ neural precursor proliferation is thought to be caused by a nicotine-induced increase in fibroblast growth factor 2 (FGF-2) mRNA in the SVZ (Figure 1). Using in situ hybridization for both FGF-2 and fibroblast growth factor receptor 1 (FGFR-1) mRNA in the SVZ, they found that 4 h after acute intermittent nicotine treatment FGF-2 mRNA was upregulated in the SVZ while FGFR-1 mRNA was not. Additionally, the use of a specific monoclonal antibody against FGF-2 administered into the lateral ventricles 24 h after nicotine treatment, blocked the enhancing effects of nicotine on SVZ neural precursor cell proliferation. Administration of a FGFR-1 inhibitor, SU5402, into the lateral ventricles 24 h after nicotine treatment also inhibited nicotine-induced effects on precursor cell proliferation [60]. Moreover, a significant reduction of precursor cell proliferation was observed when SU5402 was intraventricularly injected in control rats as well, suggesting a role for endogenous FGF-2 in normal adult SVZ neurogenesis. In order to verify that augmentation of SVZ neural precursor cell proliferation by acute nicotine treatment was mediated by activation of nicotinic receptors, pre-treatment with nonselective nAChR antagonist MECA was injected 30 min prior to acute intermittent nicotine treatments. MECA pretreatment did counteract the enhancing effects of acute intermittent nicotine on precursor cell proliferation. Further, treatment with MECA alone did not alter precursor cell proliferation in the SVZ compared to control animals [60].

Mudò et al., [60] also sought to identify cell types expressing FGF-2 and FGFR-1 in the SVZ as well as determine nicotine’s effect on precursor cell differentiation. Double-labelling of FGF-2 or FGFR-1 in combination with nestin revealed that along the SVZ ventricular epithelium only FGFR-1 was co-expressed with nestin. This suggests that SVZ precursor cells may respond to FGF-2 produced and released by other neural cell types present in and around the SVZ (Figure 1). Double-labelling of FGF-2 with GFAP demonstrated that in the SVZ FGF-2 is expressed in cells also expressing GFAP. Because both astrocytic cells and NSCs within the SVZ are GFAP-positive (as they advance down the neural lineage, NPCs and neural precursors no longer produce GFAP), they examined if Brdu+ cells were also GFAP or nestin-positive. Brdu-labeled cells were GFAP-negative, indicating nicotine-induced enhancement of proliferation in the SVZ was not targeting stem cells (SCs). On the other hand, all Brdu+ cells were nestin+ cells, signifying that actively proliferating cells are likely the target for nicotine’s proliferative effects (Figure 1). These results suggest that nicotine induces release of FGF-2 from GFAP+ cells that target FGFR-1 on nestin+ actively proliferating cells [60].

Lastly in this investigation, rats were treated with acute intermittent nicotine or saline and after 24 h received i.p. Brdu. Rats were sacrificed 12 days following nicotine treatment [60]. This resulted in all Brdu-labelled cells residing in the OB with few in the RMS and none in the SVZ. Using NeuN- and GFAP-labelling and co-localization with Brdu, no nicotine effects on differentiation of SVZ NPCs were detected. All Brdu+ cells that had migrated to the OB were NeuN+ in both saline and nicotine treated groups, and no co-localization of Brdu and GFAP were observed in either treatment group. Remarkably, in all experiments performed, the nicotine enhancement of precursor cell proliferation was not accompanied by an increase in number of apoptotic cells, as shown by comparable TUNEL-labelling in every experimental and control group. Therefore, this study indicates nicotine is not involved in mechanisms underlying fate specification of SVZ neural precursor cells nor does it influence neural precursor cell or their progeny’s survival [60].

In another study that reported a nicotine-induced increase in neurogenesis, Cohen et al., [61] demonstrated nicotine dependence and deprivation increases levels of immature neurons in the SGZ of the DG. Here, the authors also observed that nicotine positively correlates with the number of immature neurons in the hippocampus following extended nicotine access in rats. Male Wistar rats received jugular catheter implants for nicotine IVSA or did not undergo intravenous surgery to serve as controls (control animals did not receive saline/vehicle implants). Nicotine infusion were contingent upon nose-poke of active lever in nicotine IVSA group. A periodic deprivation model was used, in which weekly nicotine self-administration was available for either extended access (21 h per day) or limited access (1 h per day) for 4 d, followed by 3 d of abstinence. Nicotine self-administration continued under this model for 14 weeks and then brains were immunohistochemically analyzed. Proliferation marker, Ki-67, and immature neuron marker, neurogenic differentiation factor (NeuroD; transiently expressed in differentiating progenitors between day 1 and 7 after cell birth [62]), were not altered by limited access nicotine self-administration. While extended access nicotine self-administration and deprivation led to significantly higher numbers of Ki-67-labelled cells and NeuroD immunoreactive cells in the SGZ as compared to naïve controls [61]. A significant increase in NeuroD+ cells was observed in the SGZ of extended access rats in both pre-deprivation and post-deprivation stages as compared to controls and limited access rats. Neither limited nor extended access nicotine self-administration and deprivation was shown to induce apoptosis, as the number of activated caspase 3 labelled cells was not altered in these groups compared to naïve animals. Additionally, gliogenesis in the medial prefrontal cortex (mPFC) was evaluated and neither nicotine self-administration groups showed a difference in the number of proliferating progenitors and premyelinating oligodendrocytes. Limited- and extended-access nicotine self-administration did not alter Ki-67-labelling in the mPFC and did not change the number of Oligo2 labeled premyelinating oligodendrocytes in the mPFC compared to control animals [61].

Cohen et al., [61] revealed that in limited access rats, no significant correlation was observed between number of NeuroD+ cells and number of nicotine lever presses following depravation of nicotine. Alternatively, there was a strong correlation between number of NeuroD-immunoreactive immature neurons and the number of nicotine lever presses post-deprivation during the last week of self-administration in the extended access rats. Consequently, nicotine dependence may have negative effects on the hippocampus leading to aberrant hippocampal neurogenesis. Abnormal proliferation within the SGZ may contribute to maladaptive addiction-like behaviors that are dependent on the hippocampus. This study corroborates clinical studies that have demonstrated that nicotine-dependent individuals show pronounced impairments in hippocampal-associated behaviors such as learning and memory following abstinence from smoking [33,34,35,63,64,65,66].

## 4. Nicotine Attenuates Aß-Induced Neurotoxicity in NSCs

Evidence supports nAChRs’ role in the differentiation, maturation, survival and integration of newly born neurons in the adult brain [67,68,69]. Immature neurons derived from ongoing neurogenesis within the DG contain α7 nAChRs and receive direct cholinergic innervation that is critical for integration of these adult-born neurons into the DG network [67]. Nicotine has demonstrated neuroprotective abilities upon binding nAChRs on numerous cells throughout the brain, acting therapeutically by inducing defense mechanisms against pathology causing Alzheimer’s disease (AD) [70,71], Parkinson’s disease (PD) [20,72,73,74,75,76] and other neuroinflammatory conditions. Nicotine’s actions on hippocampal α7 nAChRs has been shown to prevent synaptic impairment induced by Aß oligomers through activation of phosphatidylinositol-3-kinase (PI3K) signaling pathways [77] (Table 3). Through its action on astrocytic α7 nAChRs and subsequent PI3K/Akt signaling transduction, nicotine protected against Aß aggregation by modulating levels of endogenous astrocytic αB-Crystallin [78]. Nicotine has been shown to protect against neuroinflammation that plays a causative role in post-operative cognitive dysfunction (POCD) [79]. Nicotine mitigated POCD in partially hepatectomized rats by reducing inflammatory cytokine expression and activating brain-derived neurotrophic factor/tropomyosin receptor kinase B (BDNF/TrkB) signaling, thus preventing neuronal apoptosis in the hippocampi of these animals [23]. Nicotine also regulates inflammation via α7 nAChRs on microglia and activating intracellular signaling cascades that ultimately inhibit the production of proinflammatory factors like TNF-α, interleukin (IL)-1 and reactive oxygen species (ROS) (for review on α7 nAChRs immune modulation see [80]). However, less is known about nicotine’s contributions, directly or indirectly, to NSC fate against neurotoxic compounds involved in such neurodegenerative and/or neuroinflammatory states.

Two in vitro studies sought to assess the impact of nicotine on Aß-induced neurotoxicity on NSCs through nicotine’s effect upon binding microglial α7 nAChRs [27,81]. The rationale being that following deposition and accumulation of Aß, a secondary phenomenon of inflammation contributes to the loss of cholinergic neurons characteristic in AD pathophysiology [82]. It is also known that Aß interacts with nicotinic receptors [83,84] and stimulating nAChRs increases Aß internalization and prevents aggregation [85,86]. Additionally, microglia play an important role in regulation of neurogenesis [87]. Taken together, cholinergic drugs may promote recovery in AD, at least partially, by restoring NSC populations, as activated nAChRs can improve the microenvironment of the brain by preventing production of microglia-derived inflammatory factors and increasing Aß phagocytosis.

The aim of the first study was just that, to improve the survival microenvironment of NSCs co-cultured with microglia by attenuating microglia-derived inflammatory factors mediated by Aß peptide accumulation [27]. To evaluate this, Jiang et al., [27] used four experimental groups: (1) a control group in which NSCs were cultured with no interventions; (2) Aß-treated group in which NSCs were exposed to 10 µmol/L of Aß_1–42_; (3) co-cultured group in which NSCs were cultured in a transwell system with microglia which had been treated with 10 µmol/L of Aß_1–42_; and (4) a nicotine pretreated group in which microglia were pretreated with 10 µmol/L nicotine for 1 h before being treated with 10 µmol/L of Aß_1–42_ and subsequently co-cultured. They found that low concentration nicotine decreased release of pro-inflammatory cytokines stimulated by Aß in primary microglia. Levels of TNF-α and IL-1 were significantly elevated in microglia co-culture group versus NSC-only control group, and these levels were decreased when treated with nicotine (10 µmol/L). Proliferative rates of NSCs were shown to be lower in Aß-treated group and even greatly diminished in the co-cultured group. Nicotine pretreatment was able to partially rescue NSC proliferative rates. Immunoreactivity for microtubule-associated protein 2 (MAP2), a neuronal marker particularly enriched in the dendrites [88], and choline acetyltransferase (CHAT), used to label cholinergic synapses [89], revealed nicotine mitigates changes in NSC differentiation provoked by microglia-derived factors induced by Aß. The control group contained 14.6% MAP2+ cells and 4.0% CHAT+ cells. While the Aß-treated group had 10.2% MAP2+ cells and 3.0% CHAT+ cells, and the co-cultured group had 5.6% MAP2+ and 0.9% CHAT+ cells. The nicotine pretreatment group was observed to have partially restored to a condition that resembled cultures without microglia present (8.3% MAP2+ cells and 2.1% CHAT+ cells). Using Annexin V/PI staining and TUNEL assays, the apoptosis rate of each condition was determined. Again, the greatest effect of Aß was seen in the presence of microglia, with the percent of apoptotic cells out of total cell population increasing from less than 10% in control to 41.3% in co-culture group. The effect of nicotine on NSC apoptosis also demonstrated the same trend as other series in this study. The apoptosis rate decreased to 29.7% but was still greater than control or Aß groups that did not contain microglia. Lastly, the authors examined the Wnt/ß-catenin pathway as the mechanism underlying the observed neuroprotective effects of nicotine. The results demonstrated Axin2 played a negative role in the Wnt/ß-catenin pathway in NSCs co-cultured with Aß-treated microglia. Nicotine enhanced Wnt/ß-catenin signaling by upregulating ß-catenin, phosphorylating glycogen synthase kinase 3ß (p-GSk-3ß), and downregulating Axin2 and phosphorylated ß-catenin. Altogether this partially ameliorating the microglia-activated inflammatory response in NSCs. These data suggest the Aß may have some direct negative consequences on NSC proliferation, differentiation and survival. However, Aß toxicity is primarily mediated through microglial actions.

The second study also used a transwell system to evaluate the indirect cytotoxicity of Aß-mediated microglial activation on NSCs [81]. Because many of the microglia-derived proinflammatory factors have been shown to induce neuron death often through mitochondrial dysfunction [90,91,92,93,94], Chen et al., sought uncover the mechanism interconnecting mitochondrial function and inflammation-mediated NSC death and how nicotine might therapeutically intervene through its actions on microglial nAChRs. In addition to evidence that the mitochondria is a primary target in inflammation-mediated neural cell death, it has also been shown that Aß progressively accumulates in the mitochondria of cells throughout the brain [95]. They found that blocking the mitochondrial permeability transition pore (mPTP) (which plays a central role in apoptotic neuronal death [96]) and activating the α7 nAChRs on microglia attenuated Aß-induced neurotoxicity on NSCs (Figure 2).

Interestingly, Chen et al., [81] did not detect α7 nAChR immunoreactivity on NSCs. This finding is in opposition to previous work that reported α7-nAChRs are present on undifferentiated stem cells and progenitor cells throughout the body [97]. Still yet, others have shown the importance of α7 nAChRs present on hippocampal NPCs and immature neurons on differentiation, maturation, integration and survival [67,98]. An exact expression pattern for nAChRs on NSCs and their progeny has not yet been clearly established. Although it appears nAChRs are vital for certain stages of neurogenesis, in this study α7 nAChR subunit mRNA was only detected on primary microglia, while no detectable expression of α7 nAChR subunit in primary NSCs was observed.

The results of this study [81] showed blocking the mPTP in NSCs or activating the α7 nAChR in microglia rescued NSC mitochondrial activity. The NSC mitochondrial membrane potential (MMP) and morphological characteristics were respectively measured by JC-1 staining and transmission electron microscopy. The addition of Aß-treated microglia more significantly altered mitochondrial depolarization in NSCs than the addition of Aß directly to NSC culture without microglia present. When Aß-treated activated microglia were pretreated with 1 µM nicotine, their negative effects on NSC mitochondria were partially reversed. The altered NSC mitochondrial membrane permeability was also partially corrected when NSCs were treated with cyclosporine A (CsA), which targets cyclophilin D, an essential regulator of the mPTP opening [99], and ultimately blocked the mPTP. Nicotine or CsA pretreatment was also shown to reverse Aß-induced mitochondrial cristae swelling. Lastly, Aß-mediated microglial activation greatly increased apoptotic rate of NSCs compared to control. Nicotine activation of microglial α7 nAChRs and blocking of the mPTPs by CsA attenuated the observed NSC apoptosis.

## 5. Nicotine Induces Mitochondrial Stress in Neural Stem Cells

Changes to SC mitochondrial health could have highly important implications as mitochondria play a key role in regulating stemness [100,101,102,103,104,105,106]. Recently, in addition to preventing senescence, mitochondria have also been linked to stem cell activation and determining fate/differentiation [105,106,107]. Unfortunately, mitochondria are also sensitive to stress [108], leaving them susceptible to possible damage from nicotine and/or other toxic byproducts from tobacco use. Moreover, a decline in SC mitochondria function could underlie accelerated aging traditionally seen in smokers.

Studies have suggested nicotine exposure effects the mitochondrial respiratory chain, oxidative stress, calcium homeostasis, mitochondrial membrane proteins, mitochondrial association from microtubules, mitochondrial dynamics, biogenesis, and mitophagy [109,110] (for full review on nicotine’s impact on mitochondrial activity see [111]). However, few studies have specifically examined the impact of nicotine exposure on NSCs and their mitochondria. As discussed in the previous section (Nicotine attenuates Aß-induced neurotoxicity in NSCs), Chen et al., demonstrated a link between NSC mitochondrial function and Aß-induced neurotoxicity. Interestingly, the results of that study advocate for a protective role of nicotine against microglia-derived proinflammatory factors that have been shown to induce neuron death via mitochondrial dysfunction (Figure 2). In a transwell coculture system, Aß-mediated microglial activation led to mitochondrial dysfunction within NSCs and caused NSC apoptosis. These negative effects were attenuated by treatment of microglia with 1 µM nicotine and treatment of neurospheres with a mitochondrial permeability transition pore inhibitor. This suggests that mitochondria may play a critical role in nicotine’s action as a neuroprotective agent for NSCs.

In contrast, Zahedi et al., demonstrated a mitochondrial stress response in NSCs following exposure to electronic cigarette ENDS e-liquids and their aerosols [112]. Importantly, they also reported that the effects of ENDS on the mitochondria are mediated by nicotine and not by the transfer of volatile organic chemicals or solvents found in e-liquids (propylene glycol and glycerin). Cultured NSCs were incubated for 4 or 24 h with menthol- or tobacco-flavored e-liquids (0.3%, 0.5% and 1% dilutions) or aerosols (1, 3, and 6 total-puff-equivalents (TPE)). The results of this study revealed that EC liquids and aerosols inhibit proper autophagy in NSCs. Autophagosomes were enlarged in a time-dependent manner, with a greater enlargement seen in e-liquid exposed than aerosol exposed cells. The enlarged autophagosomes also demonstrated an increase in pH. (Figure 2). The decrease in autolysosome acidity could decrease its normal proteolytic function. This is important as mitophagy, or mitochondrial autophagy, protects cells by removing damaged mitochondria which could induce pro-apoptotic signaling. After 4 and 24 h e-liquid and aerosol treatments mitophagy had not significantly increased, except in the high-dose 1% menthol-flavored nicotine e-liquid group. Using time-lapse imaging and a motion-magnification algorithm they did observe e-liquids and aerosols alter mitochondrial dynamics. Treatment with both e-liquids decreased large motion and increased small motion as compared to controls. While menthol-flavored aerosol decreased small motion and increased large motion of NSC mitochondria, and the same trend was observed in tobacco-flavored aerosol treatment but was not statistically significant when compared to control.

The changes observed in mitochondrial dynamics appear to correlate with alterations in mitochondrial swelling or hyperfusion (Figure 2). Greater motion was seen in hyperfused mitochondria (aerosol) and less motion seen in the swollen mitochondria (e-liquids). As hyperfusion has been shown to increase superoxide production [113], they loaded cells with a mitochondrial-targeted dye that produces fluorescence in the presence of superoxide anion. A dose-dependent escalation of fluorescent intensity was observed in e-liquid-treated cells, and a lesser increase also found in the aerosol-treated cells. Using immunoblot analysis, it was determined that cells treated with e-liquid or aerosol also had significantly elevated superoxide dismutase two levels compared to controls. Additionally, a concentration-dependent increase in mitochondrial protein oxidation was observed in all e-liquid and aerosol treatments.

This study also provided evidence that EC e-liquids and aerosols can cause significant, not-easily-reversible damage to the mitochondria of NSCs. EC treatments led to altered MMP, induced aggregation of mitochondrial nucleoids (Figure 2) and mitochondrial DNA (mtDNA), and induced calcium influx (Figure 2) leading to plasma membrane retraction and intracellular calcium overload. Labelling with tetramethylrhodamine methyl ester (TMRM) dye, a cell-permeable, cationic dye that is sequestered by active mitochondria but leaks out of depolarized mitochondria, revealed changes in MMP in NSCs treated with high-dose e-liquid or aerosol for 24 h. Treatment with aerosols (hyperfused mitochondria) increased TMRM accumulation in the mitochondria. While 1% liquid (swollen mitochondria) had different results. The 1% tobacco-flavored nicotine e-liquid showed no change in TMRM, and the 1% menthol-flavored nicotine e-liquid dose showed a decrease in TMRM signal, suggesting this concentration damages the mitochondria and causes membrane leakage. Because an elevation of ROS can lead to mtDNA damage, a dye was used to label mitochondrial nucleoids. Control cells had multiple small fluorescent nucleoids, whereas e-liquid (0.5%) and aerosol (6TPE) for 24 h treated cells had larger, brighter nucleoids characteristic of mtDNA aggregation.

Lastly, EC- and nicotine-induced alterations to calcium influx were evaluated. NSCs transfected with a fluorescently tagged calcium reporter (GCaMP5) and imaged live before and after exposure to e-liquid (0.5% menthol- or tobacco-flavored), aerosols (6TPE menthol- or tobacco-flavored), and various concentrations of nicotine (220 µg/mL or 1.1 µg/mL). Increase in intensity was seen at one minute for all conditions and was most pronounced in 0.5% menthol-flavored nicotine e-liquid and 220 µg/mL nicotine solution. These data also showed that calcium levels did not return to baseline by 20 min after exposure, resulting in calcium overload. Much of the fluorescent signal was seen accumulating in the perinuclear region, which was presumed to be due to sequestering of calcium by the endoplasmic reticulum. However, because intracellular calcium can also accumulate in the mitochondria which can then trigger ROS production [114], NSCs were then transfected with a genetically encoded calcium indicator called calcium-measuring organelle-entrapped protein indicator (CEPIA). Transfected cells were again imaged live before and after exposure to e-liquid (0.5% menthol- or tobacco-flavored), aerosols (6TPE menthol- or tobacco-flavored), and various concentrations of nicotine (220 µg/mL or 1.1 µg/mL). All treatments showed an increase in fluorescent signal within 1 min, followed by a decrease by 44 min. Again, the greatest changes were observed 0.5% menthol-flavored nicotine e-liquid and 220 µg/mL nicotine. Unlike the other treatment, the calcium levels in these groups did not return to resting state by 20 min following exposure. NSCs were then pre-incubated with Ru360, a compound to block calcium uptake through mitochondrial calcium uniporter (MCU) proteins. With calcium uptake through MCU inhibited, the changes in intracellular calcium following e-liquid, aerosol, or nicotine exposure were significantly dampened. Mitochondrial calcium influx was also successfully inhibited by blocking α7 nAChRs with α-bungarotoxin (α7 nAChR competitive antagonist). Lastly, they sought to evaluate if calcium blockage could impede mitochondrial effects of ECs and nicotine. Thus, NSCs were incubated with 1 µM calcium chelator EGTA (ethylene glycol-bis(ß-aminoethyl ether)-N,N,N’,N’-tetraacetic acid) and treated with 0.5% e-liquids for 4 h. The results showed a significant decrease in mitochondrial superoxide production when EGTA was present. The authors conclude these data suggest calcium overload contributes to EC-induced mitochondrial defects in NSCs.

## 6. Discussion

Adult neurogenesis plays an important role in learning and memory [115], olfaction [12], and the brain’s ability to regenerate after injury [116]. Albeit limited, the brain does have some capacity to self-heal following insult such as stroke. Ischemic and traumatic lesions within the cortex have been shown to stimulate proliferation in the SVZ [117,118,119,120,121]. These newly born neural cells then migrate along blood vessels [122] to the site of injury to where they differentiate primarily into glial cells and fewer become mature neurons [117,120,123]. One physiological limitation is a substantial lack of integration of these recruited endogenous precursor cells, particularly in the case of stroke as ischemia creates a very hostile microenvironment for the newly arriving cells [123]. Since the revelation that neural precursors can migrate to sites of injury within the brain, considerable research efforts have been placed on creating a more robust neurogenic response, promoting NPC migration and enhancing integrative capacity into damaged regions following stroke and other neurodegenerative conditions [124,125,126,127,128,129,130]. As this field of regenerative research advances, it will be important to understand what effects the main risk factors for stroke, such as smoking, have on the health of NSCs.

In addition to effects nicotine may have on continued postnatal neurogenesis and its implications for the possible exploitation of NSCs therapeutically in adults, it is also important to consider nicotine-induced changes in prenatal neurogenesis. While the scope of this review primarily focuses on SCs and progenitors of adult neurogenesis, it is important to note that ENDSs are often used by adolescents and pregnant women. During development, SCs of the nervous system are especially vulnerable to toxicants [131]. In attempts to abandon smoking, pregnant women may resort to the use of nicotine patches or ENDSs that are colloquially considered less dangerous [132]. These methods nevertheless introduce nicotine, which is considered a neurotoxic and neuroteratogenic agent [133,134,135,136], to the developing fetus. Notably, evaluation of maternal nicotine exposure during gestation and lactation period in mice revealed alterations in neurogenesis in offspring [137]. Nicotine-exposed offspring were found to have altered behavior, increased cell proliferation in the DG, more disorganized immature doublecortin positive (DCX+) neurons and increased microglial markers.

The mRNA for neuronal nicotinic receptor subunits and nicotine receptor binding sites are first detected at the end of neurulation, and at that point the developing brain becomes vulnerable to nicotine exposure [138,139]. Importantly, this process occurs early in pregnancy as primary neurulation begins week 3 of gestation [140]. Newly born cells destined for neuronal lineage migrate from the mitotic zone to various regions of the developing brain then nAChR expression becomes regionally specific [141,142]. As previously mentioned, nAChRs play a role in differentiation, maturation, integration, and survival of newly born neurons throughout adulthood [67,68,69]. In the fetal brain acetylcholine signaling is important for coordinating proper assembly of the neural tissues [143]. Maternal cigarette smoking has been associated with structural and functional alterations within the brain, including in the mPFC [144], striatum, amygdala, and hippocampus [145], and numerous cognitive and behavioral changes [146,147], such as deficits in learning and memory [148], auditory dysfunction [136], hyperactivity [149], and depressive-like behaviors [150], in the offspring (for a full review of pre-clinical and clinical evidence of nicotine’s effects on development in the nervous system and more see [151]).

## 7. Conclusions

The current state of research on nicotine’s effects on neural stem, progenitor and precursor cells is still in its infancy and there is much to be discovered. Presently nicotine appears to have differential effects on these immature cells of neural lineage depending on both location and maturation. Nicotine may exert a protective role for NSCs when exposed to cytotoxic or inflammatory agents. The mitochondria of NSCs appear to also play an important role in these interactions between NSCs, glia, and nicotine. Additionally, nicotine may have differing effects on NSCs depending on route of administration and further research is needed to determine these consequences.

## Figures and Tables

**Figure 1 brainsci-11-00172-f001:**
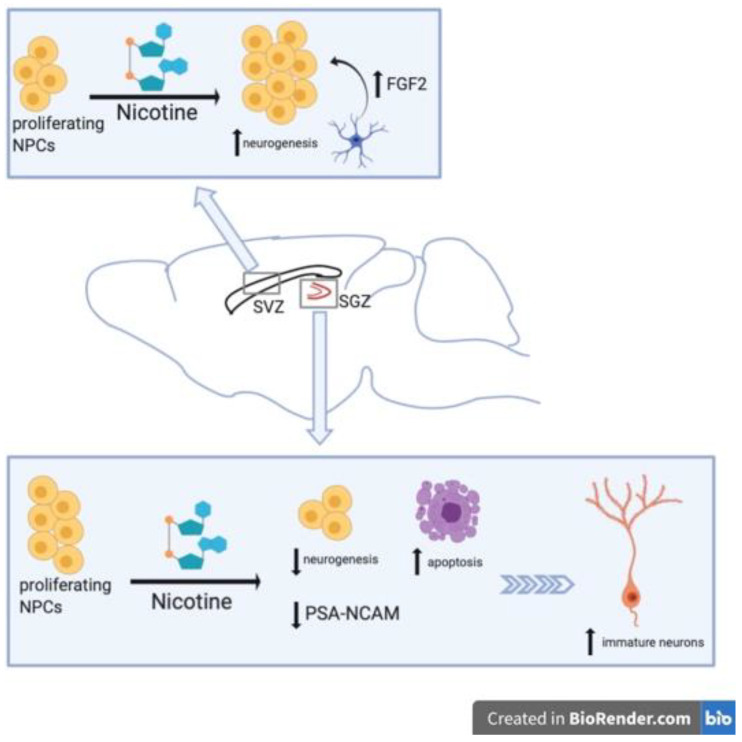
Graphical summary of nicotine’s effects on NSC proliferation. Nicotine exhibits region-specific effects on neurogenesis. In the subventricular zone (SVZ), nicotine exposure enhances neurogenesis while in the SGZ it decreases neurogenesis.

**Figure 2 brainsci-11-00172-f002:**
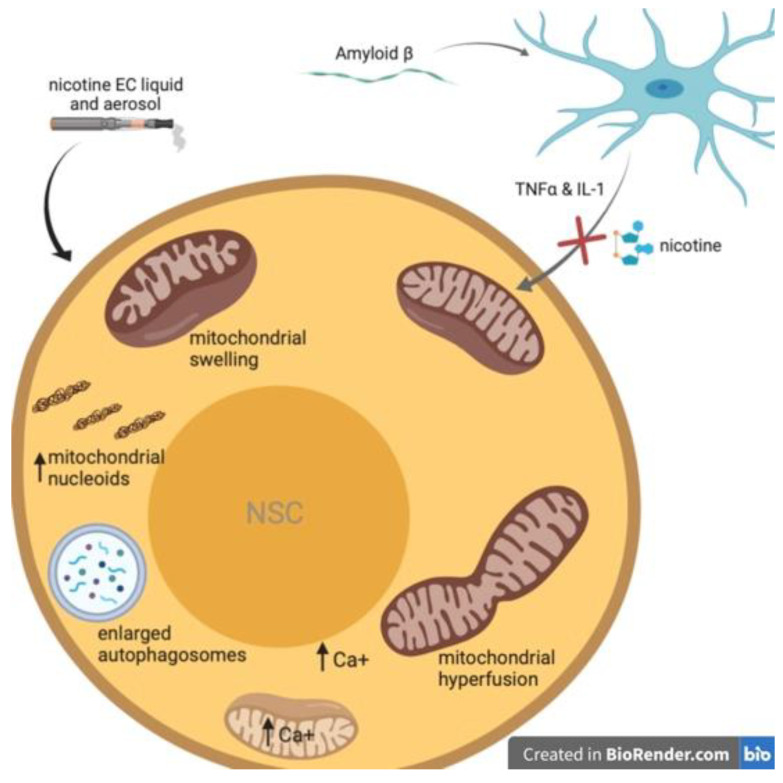
Summary of described effects of nicotine e-cigarettes (EC) liquid and aerosols on NSC mitochondria. EC treatments led to altered mitochondrial membrane potential (MMP), induced aggregation of mitochondrial nucleoids and mitochondrial DNA, and induced calcium influx leading to plasma membrane retraction and intracellular calcium overload.

**Table 1 brainsci-11-00172-t001:** Summary of nicotine-related decreases in neural stem cells (NSC) proliferation.

Observation	Nicotine Doses	Model System	Ref
Nicotine reduces proliferation of NSCs via transcriptional mechanisms	100, 400, 800 µM	Neurospheres	[40]


Nicotine reduced the NSC proliferation in the dentate gyrus (DG)	4 mg/kg/day	Mice	[52]
Nicotine reduced the NSC proliferation in the DG	Variable (IVSA)	Rats	[55]
Nicotine decreases hippocampal neurogenesis	0.1, 0.5, or 1 mg/kg	Rats	[53]



**Table 2 brainsci-11-00172-t002:** Summary of nicotine-related increases in NSC proliferation.

Observation	Nicotine Doses	Model System	Ref
Intermittent nicotine exposure increases neural precursor proliferation in the SVZ due to increases in FGF-2	1 mg/kg	Rats	[60]


Nicotine dependence and deprivation increases the number of immature neurons in the SGZ of the DG	Variable, IVSA.	Rats	[61]


**Table 3 brainsci-11-00172-t003:** Nicotine-related neuroprotection and neurotoxicity in NSCs.

Observation	Nicotine Doses	Model System	Ref
Nicotine prevents Aβ aggregation via hippocampal α7 nAChRs	10 µM (cultures), 1 mg/kg (mice)	Cultured hippocampal neurons, mice	[77]
Nicotine is neuroprotective by reducing inflammation by activating BDNF/TrkB and inhibiting TNF-α and IL-1	0.5 mg/kg	Rats	[79]
Nicotine-induced activation of α7 nAChRs attenuates Aβ-induced neurotoxicity by reducing Aβ accumulation in mitochondria	1 µM	cultured hippocampal NSCs	[81]

## Data Availability

No new data were created or analyzed in this study. Data sharing is not applicable to this article.

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
