# Peer review of "Systematic Review of Nicotine Exposure’s Effects on Neural Stem and Progenitor Cells"

_brainsci, 2021, doi:10.3390/brainsci11020172_

Round 1
Reviewer 1 Report
This is an extensive and thorough review of effects of nicotine exposure on neural stem/progenitor cells (NSC). Considering the high smoking rate in both adult and middle/high school students, it is of great importance to understand the potential impacts of nicotine exposure on neural stem/progenitor cells. The authors reviewed studies on nicotine-related decreases and increases in NSC proliferation, Aß-induced neurotoxicity attenuated by nicotine, and nicotine induced mitochondrial stress in NSC. Nicotine appears to have differential effects on those immature cells of neural lineage depending on the location and maturation. This review may shed light on future studies in terms of figuring out the interaction between NSCs, glia and nicotine.
Author Response
We appreciate the fact that Reviewer 1 rated our initial version of the review so well.
Reviewer 2 Report
This is an intensively-researched review article on the nicotine exposure’s effects on neural stem and progenitor cells. While the entire review article is easy to understand, there are some still some grammatical errors in several sections. I highly recommend that the authors have the manuscript proofread. Some minor comments that I have:
1. Please incorporate one summary table each for sections 2,3,4. Because there are too much evidence presented in the text, having a summary table will allow readers to have a quick glance on the evidence on the effects of nicotine without getting lost when reading through the texts.
2. All figure legends should briefly describe the diagram, not just merely stating what the figure is.
3. The in-text citation such as Chen, et al. should be written as Chen et al. Please remove all the comas following the last name of the authors.
4. Please make sure all the abbreviations used are kept consistent throughout the texts. For instance, use either hours or h.
5. Please check the format of your references. Some page numbers
Author Response
We thank the reviewer for their time and consideration. We have fixed numerous grammatical issues in this revised version of our review. To address the individual comments:
- We have now included a summary table for sections 2, 3, and 4 as suggested.
- All Figure legends include brief summaries instead of just headings.
- In-text citations have been reformatted.
- Abbreviations have been updated to be more consistent
- We have checked the format of our references with attention to page numbers. Many of our references lack a page number due to the online-only format of some of the journals.